# Transformation of Indonesian Health System: The Impact on Medical Education

**Gilbert Sterling Octavius** [1,*] **, Rhendy Wijayanto** [2] **and Theo Audi Yanto** [3]

1   Department of Pediatrics, Universitas Pelita Harapan, Tangerang 15811, Indonesia
2   Department of Medical Education, Universitas Pelita Harapan, Tangerang 15811, Indonesia
3   Department on Internal Medicine, Universitas Pelita Harapan, Tangerang 15811, Indonesia
*   Correspondence: sterlinggilbert613@hotmail.com

**Abstract:** Indonesia is currently revamping its medical sector—a process dubbed medical transformation. In place of this transformation, medical education has also received spotlights due to the number of medical universities and the lack of medical specialists in Indonesia. Therefore, several plans will revolutionize Indonesian medical education and its health system. This commentary will briefly comment on those transformations and their potential impact in the near and distant future.

**Keywords:** Indonesia; transformation; health system; medical education

"It is a simple production issue, not a distribution one"—as stated by the Indonesian Minister of Health in one online meeting session.

Indonesia is currently undergoing a public health transformation. In November 2022, the Indonesian Minister of Health announced six foundational pillars that will be the basis of this massive project, such as the transformation of primary health care, the transformation of referral care, the transformation of the health resilience system, the transformation of healthcare funding, the transformation of healthcare technology, and finally, the transformation of human resources in healthcare [1]. In this commentary, we will discuss the last pillar in more depth.

Amongst the plans for transforming human resources in healthcare are increasing the quota for medical student admissions, offering more medical scholarships within and outside of Indonesia, and streamlining the process of foreign doctors and Indonesian doctors who studied abroad to practice clinical medicine in Indonesia. The rationalization behind these steps is the perceived lack of medical doctors, general practitioners (GP) and specialists, and the distributional approach.

Hence, the Indonesian Minister of Health is collaborating with the Minister of Education to tackle this issue. The more highlighted problem is the lack of specialists, namely, internists, pediatricians, obstetrical oncologists, and cardiologists, to name a few. Currently, Indonesia adopts a university-based approach, where GPs who want to become specialists must enroll as students in one of the 21 medical universities that can carry out specialistic education. However, not all universities can open all specialistic divisions; hence, there are currently 263 specialist study programs out of the possible 735 programs in all universities [2].

The lack of production capacity is seen as a bottleneck for specialist production. Therefore, the Minister of Health proposes implementing a hospital-based approach to rapidly increase the number of specialists. A hospital-based system may not be an issue in other countries, but this has been unheard of in Indonesia until now. While entering these uncharted territories, many questions and much confusion regarding medical curriculum, legal issues, responsibilities of all stakeholders, and ensuring education quality has ensued. This quantity-versus-quality debate sparked because Indonesia has adopted an accreditation approach to ensure quality. There is a tangible proxy for measuring quality, as

Indonesia is still trying to ascertain how to measure quality in a hospital-based approach setting. The accreditation status grades a university from A (excellent) to C (good enough). Only universities with an "A" accreditation status may open a specialistic education, which is 21 out of 93 universities. There are currently 100 more universities waiting for approval to open a medical faculty. Hence, some sentiments are shared amongst clinicians to increase universities' quality so they can open a specialistic course rather than shifting into a hospital-based approach suddenly.

The university's role during the transition from a university-based to a hospital-based system may be underappreciated. Universities may act as a bridge to support and facilitate the evolution and development of medical curricula for future residents in a hospital-based system. Aspiring residents in a future hospital-based program can benefit from excellent experiential learning opportunities that prepare them for issues in the real world by integrating medical curricula from universities with hospital-based training. Establishing collaborations between universities and hospitals, including clinical rotations in different hospitals, and encouraging multidisciplinary cooperation are just a few strategies that can help with this transformation. Future healthcare professionals will be well-equipped to meet the changing requirements of patients and healthcare systems because of this convergence of academia and clinical practice [3].

The not-so-popular topic is how this transformation will affect the medical curriculum for future GPs. While equipping more primary health care (PHC) with essential and state-of-the-art equipment such as ultrasonography, mammography, and genomic and genetic screening, GPs must be equipped with the knowledge. In our opinion, the current medical curriculum does not educate and prepare future GPsto interpret cancer screening results, which may lead to overdiagnosis and overtreatment. While there is no concrete data on the statistical literacy of Indonesian medical workers, studies from other countries with a more robust research background and better statistical literacy show that GPs still make lapses in clinical judgments and errors while interpreting screening results. Hence, without proper knowledge and experience, patients will ultimately be confused, leading to a lack of trust in medical workers [4]. The current medical curriculum does not equip medical students with the basics of genomics and genetic screening, when to order such tests, interpret them, and act based on the results. Basic literacy definitely will not be adequate to prepare GPs in clinical settings [5]. One may also argue that the downside to a university-based approach is a continuous emphasis on education focused on specialists at the expense of family medicine and public health. A more public-health-centric education needs more attention than ever if the Minister of Health wants all six pillars of transformation to be successful.

Some positive progress is being made in advancing public-health awareness and education in GPs in Indonesia. One study conducted at Universitas Gadjah Mada, Yogyakarta, qualitatively surveyed 19 GPs' perspectives on formal postgraduate training in primary care. The results showed that the primary care training curriculum was unfamiliar to almost all GP participants. General practitioners also doubted that the training could alter the nation's health care system given its lack of facilities and funding. However, exposure to the training produced encouraging indications that it would enhance the physicians' expertise in primary care practice [6], Another study aimed to evaluate a graduate professional training program in general practice using the "experiential learning" concept. Still based in Universitas Gadjah Mada, the study concluded that a framework for experiential learning helps improve general practitioners' understanding of better primary care practice [7]. These two papers provide an evidence for transforming primary health care in Indonesia through medical education. Indonesian biomedical literacy and research are still far below standard [8]. Numerous studies have shown that GPs and residents have low-to-below-average statistical literacy [9]. Statistical illiteracy will translate into different interpretations of the same result, confusing the patients or harming them. This skill is crucial if cancer screening is to be transformed in Indonesia. All general practitioners must understand the concepts of sensitivity, specificity, numbers needed to treat, likelihood ratios, and other terminologies. We argue that biomedical literacy needs to be taught more

and with a deeper understanding in Indonesia as continuing medical education (CME) may not be enough to tackle this issue [10].

Indonesian medical education may not undergo such a massive transformation as in Vietnam, but it is still a significant shake-up. In its pre-reform clinical curriculum, Vietnam also had its share of problems. Their curriculum was heavy in traditional lectures, passive learning, teacher focused, and focused on in-hospital care. The objective was to change the curriculum from one that was discipline and knowledge based to one that was integrated, system-, and competency-based [11]. While Indonesia and Vietnam face different situations and challenges in the medical education context, Indonesia can benefit from the lessons learned in reforming a medical curriculum. In this commentary, we would like to point out that Indonesia's medical health system transformation should be thoroughly analyzed beyond the production issue. There are still vital challenges in Indonesia's medical education system that need solving to support the six pillars of transformation, such as: a more public-health-tailored curriculum; an updated teaching method that incorporates pedagogical competence; recruiting local medical professionals from abroad to tackle the shortage of full-time lecturers as they can speak the local language and are more aware of the cultures and inadequate curriculum to impart bioethical and biostatistical literacy; as well as ensuring appropriate accreditation and standardization of medical schools, especially if all 100 new medical faculties were to be approved [12].

**Author Contributions:** Conceptualization, G.S.O.; writing—original draft preparation, G.S.O., R.W. and T.A.Y.; writing—review and editing, G.S.O., R.W. and T.A.Y.; supervision, T.A.Y.; project administration, R.W. and T.A.Y. All authors have read and agreed to the published version of the manuscript.

**Funding:** This research received no external funding.

**Institutional Review Board Statement:** Not applicable.

**Informed Consent Statement:** Not applicable.

**Data Availability Statement:** Not applicable.

**Conflicts of Interest:** The authors declare no conflict of interest.

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
