# Peer review of "Transformation of Indonesian Health System: The Impact on Medical Education"

_ime, doi:10.3390/ime2020009_

Round 1
Reviewer 1 Report
Among Indonesia's recent public health transformation, the authors point out issues they face in educating healthcare professionals and make important recommendations for solutions.
This article shows the number of medical specialists has been in shortage in Indonesia. They shifted from conventional university-based education to hospital-based education to rapidly increase the number of specialists. However, to maintain the quality of education, the authors suggest that it is important to establish a system for evaluating and certifying the quality of specialist education.
They also point out the need for curricula and programs to educate GPs on literacy about clinical medicine and research so that they can use state-of-the-art medical knowledge and equipment.
This Communication paper is a very useful paper for educators training medical professionals in Indonesia, so I agree with its acceptance to this journal.
Reviewer 2 Report
This is a very interesting comment on Indonesia's medical health system transformation. The authors make it clear that this transformation should be analysed thoroughly beyond the production/capacity issue.
Some minor comments:
· Paragraph 4 (line 39-52), in my opinion, could improve by describing a more specific role/stronger positioning of an educational department. This role includes supporting and facilitating transitions and development in the medical curriculum including the medical specialist training institute.
· Paragraph 5 describes the medical curriculum for future GPs. Worth to mention here is a recent and promising development in this field: the start of a Family Medicine specialist programme in Indonesia, for example at Universitas Gadjah Mada and UI (Claramita et al., 2018; Ekawati et al., 2018).
· Line 56-58 starts with ‘In our opinion’, while it also refers to ref 3. Please provide some more clarity on this.
· In line 75, the Indonesian situation is compared with the massive transformation in medical education in Vietnam. For me, as a somewhat naïve reader, it is not completely clear why these two transformations are compared. Some more relevant background on this comparison could help me.
Claramita, M., Ekawati, F. M., Gayatri, A., Istiono, W., Sutomo, A. H., Kusnanto, H., & Graber, M. A. (2018). Preparatory graduate professional training in general practice by using the 'experiential learning' framework. Asia Pacific Family Medicine, 17(1), 4. https://doi.org/10.1186/s12930-018-0042-1
Ekawati, F. M., Claramita, M., Istiono, W., Kusnanto, H., & Sutomo, A. H. (2018). The Indonesian general practitioners' perspectives on formal postgraduate training in primary care. Asia Pac Fam Med, 17, 10. https://doi.org/10.1186/s12930-018-0047-9
